# The construction process for pre-stressed ultra high performance concrete communication tower

**Voo Yen Lai[1,2]**, **Farzad Hejazi** [1] *, **Sarah Saleem[1]**

**1** Department of Civil Engineering, University Putra Malaysia, Selangor, Malaysia, **2** Dura Technology Company, Chemor, Perak, Malaysia

☯ These authors contributed equally to this work.

* farzad@fhejazi.com

**Data Availability Statement:** All relevant data are within the paper.

**Funding:** This research receives support from Dura Technology Sdn. Bhd. (https://dura.com.my) via Dr. Voo Yen Lai with grant project with Vot No.

## Abstract

Towers are important structures for installing radio equipment to emit electromagnetic waves that allow radio, television and/or mobile communications to function. Feasibility, cost, and speed of the construction are considered in the design process as well as providing stability and functionality for the communication tower. This study proposes the new design for construction of segmental tubular section communication tower with ultra-high-performance fibre concrete (UHPFC) material and prestress tendon to gain durability, ductility, and strength. The proposed mix design for UHPFC in this study which used for construction of communication tower is consisted of densified Silica Fume, Silica fine and coarse Sand and hooked-ends Steel Fiber. The prestressed tendon is used in the tower body to provide sufficient strength against the lateral load. The proposed design allows the tower to be built with three precast segments that are connected using bolts and nuts. This paper presents a novel method of construction and installation of the communication tower. The advantages of proposed design and construction process include rapid casting of the precast segment for the tower and efficient installation of segments in the project. The use of UHPFC material with high strength and prestress tendon can reduce the size and thickness of the tower as well as the cost of construction. Notably, this material can also facilitate the construction and installation procedure.

## 1 Introduction

Ultra high-performance concrete (UHPFC) exhibits advantageous durability properties such as low porosity, extremely low permeability, high ductility, and resistance to leaching and corrosion. After thermal treatment, there is no additional shrinkage with very little creep is observed [1]. Many studies have been done on various materials and durability properties of UHPC [2–4].

UHPC is a new generation of engineering infrastructure material. It is a special type of concrete that has attracted many civil engineers because of its strength and durability performance. Both production and application are implemented in the latest knowledge and technology of

6300195. Dura Technology Sdn. Bhd. received this fund from Ministry of Science Technology & Innovation, Malaysia (https://www.mestecc.gov.my) for the project entitled as "Development and construction of Internet Transmission Tower Using Ultra High Performance Concrete". Their help and support are gratefully acknowledged. The sponsors or funders have no any role in the study design, data collection and analysis, decision to publish, or preparation of the manuscript.

**Competing interests:** This research receives support from Dura Technology Sdn. Bhd. (https://dura.com.my). This does not alter our adherence to PLOS ONE policies on sharing data and materials.

concrete manufacturing [5]. Those new types of structures using UHPC materials are Shepherds Creek Bridge of Australia [6], Wapello Bridge in Iowa of the United States [1], Kuyshu High Speed Bridge of Japan [7], and FHWA short span bridge of the United States [8].

There are many reports talking about the application of UHPC in hybrid beams. Yoo and Yoon [4] studied the structural performance of UHPC beams with different steel fibers. Their results indicated that steel fibers significantly improved the load carrying capacity, post-cracking stiffness, and cracking response. At the same time, steel fibers decreased the ductility. With the inclusion of 2% volume of steel fibers, it was expected to obtain a higher load carrying capacity between 27% and 54% and a lower ductility between 13% and 73%. In addition, an increase in the length of smooth steel fibers and the use of twisted steel fibers enhanced the post-peak response and ductility. However, no noticeable difference was found in the load carrying capacity and post-cracking stiffness. Cracking responded accordingly to the length and type of fiber. Farhat et al. [9] indicated that UHPFC prevented shear failure of the beams and the failure load increased up to 86%. Jabbar et al. [10] noted that the capacity of UHPC beams for twisting was twice that of HSC beams.

The connection of discrete elements in precast concrete structures is important for overall continuity of the building [11]. The economic advantages of precast concrete structures are better than in-situ concrete structures. The existing benefits include improved finishing, quality control, curing and concrete casting [12]. Singh [13] discussed the design concepts of concrete towers both in-situ and pre-cast construction. He noted that the concrete had higher material damping properties than other materials. Moreover, prestressed concrete had strong fatigue resistance with high tolerance and less risk from dynamic failure. Ali and Moon [14] demonstrated that the efficiency and economy of the tall buildings should be examined. Haar and Marx [15] discussed the methods of designing and constructing the wind turbine, concrete support structures and different concrete tower concepts. In a nutshell, using precast concrete towers design was a fast construction process on site. Abdelrazaq et al. [16] and Baker and Pawlikowski [17] showed a brief development and construction planning of super tall buildings (Burj Dubai) project.

The use of high-strength steel strands at post-tension concrete towers that increase the structure's bending strength, shear strength, resistance to fatigue effects and lateral loading. Prestressed concrete becomes more economical than steel construction. Lanier [18] evaluated the cost effectiveness of using hybrid steel/concrete or full-height prestressed concrete towers in conjunction with self-erecting schemes for large wind turbine applications as part of the Low Wind Speed Turbine (LWST) project. LaNier compared it to baseline all-steel tubular towers and found that the cast-in-place concrete tower construction approach incurred the lowest cost solution. Quilligan et al. [19] distinguished the heights between the steel and pre-stressed concrete wind turbine towers within the range from 88m to 120m. The results suggested that the prestressed concrete towers provided a viable alternative to improve the performance. Wu et al. [20] proposed a Post-tensioned Ultra High Performance Cementitious Composites with compressive strength of 200MPa for hybrid wind turbine tower by using different cases of wall thickness and pre-stressing tendons. They found out that the wall thickness could influence the displacement. The pre-stress could effectively reduce the stress of the tower to ensure a better stability. Preciado et al. [21] proposed external prestressing devices for the seismic vulnerability reduction of masonry towers. The medium prestressing level enhanced the force capacity of towers failing by pure bending without reducing the ductility. On the contrary, the high prestressing level slightly reduced the displacement capability of towers failing ductile.

Beside UHPFRC concrete material, many studies have been conducted about special high dense concrete materials to produce advanced containment for radioactive waste [22]. Saleh

et al [23] investigated on mechanical and physical characterization of cement reinforced by iron slag and titanate nanofibers [23, 24] and hard cement-recycled polystyrene composite [25] to produce advanced containment. Also influence of severe climatic variability on mechanical and chemical stability of cement kiln dust-slag-nanosilica composite used for rad-waste solidification has been investigated by Saleh et al [25, 26].

The requirement of existing communication towers is increasing for the reason for the growing demand of power supply and telecommunication services. Steel towers are featured by their lightweight structures with high slenderness and large flexibility due to some operational and economic reasons; however, the corrosion of steel always is a challenging issue for tower maintenance. The previous studies showed that Ultra-High-Performance Concrete provides great strength and less weight simultaneously. The UHPC material with strength of 200 MPa is almost similar to the steel except for its tensile capacity which is still comparatively low. Furthermore, its upgraded strength properties, its special resistance against all kinds of corrosion are another step towards no-maintenance constructions.

Therefore two main gaps and challenges which identified through extensive review of the literature regarding communication towers as demonstrated as follow:

i) The steel towers are conventional structure system which are using to construct the communication tower due to easy transportation and fast installation. However the corrosion of steel members is the most challenging issue for using steel communication towers which is leading to increase high cost of maintenance.

ii) Casting of concrete communication tower is not practical since it is required to install molding for tower and make in-situ concrete as conventional precast concrete strength is not enough to transport to site and assemble through dry joint.

Therefore, the main objective of this study is developing new method for design and construction of segmental precast communication tower by using Ultra High Performance Steel Fibre Concrete. Hence, implementing UHPFRC material for construction segmental tower is resulted to enough strength for each segment of tower with low weight (less thickness) to make feasibility to transport to the site and install it using concrete dry joint. As UHPFRC is highly resisting against corrosion, therefore the durability of communication tower in high humidity areas is noticeable increased.

Therefore, in this study an attempt has been made to develop, the design and construction procedures for segmental prestress UHPFC communication tower and the proposed design and construction procedure is implemented to build a 30 meters UHPC communication tower.

## 2 Application of UHPFC in communication tower

Concrete is a material providing strength against imposed load [13]. Since communication tower is often exposed to wind, sun, rain, and cold conditions, utilizing tailor-made concrete which meets specific requirements will guaranty the tower's durability. This study focuses on the design and construction of communication tower with UHPFC materials.

The proposed mix design for UHPFRC material which used in this study to construct the communication tower has been illustrated in Table 1.

### 2.1. UHPFC mix design

Material composition of the UHPFC matrix are Densified Silica Fume (SF90), Dry Silica Fine Sand, Dry Silica Coarse Sand, Silica, Steel Fiber and Free Water. The exact proportion of the material used for the mixing is as shown in Table 2. The mix proportion is adopted from Wille et al. [19] and target means compressive strength of UHPFRC used in the tower, was 150MPa.

**Table 1. Mix design for UHPFRC used for construction of communication tower.**

| Material | Weight (kg/m$^3$) |
|---|---|
| Cement | 850 |
| Densified Silica Fume (SF90) | 200 |
| Dry Silica Fine Sand 30/100 PB | 695 |
| Dry Silica Coarse Sand 16/30 PB | 295 |
| Silica VC2644 | 40 |
| Hooked-ends Steel Fiber | 158 |
| Free Water | 140 |
| 3% moisture | 30.93 |
| Total air voids | - |
| Total | 2408.93 |

Also, the material properties of steel reinforcement used in this research project has been presented in Table 3.

## 2.2. Mechanics behavior of UHPFC

The material properties of UHPFC is presented in Table 4 for both transferring and service stages. Also the results of compression test of UHPFC concrete in 7, 14 and 28 days are presented in Table 5.

Both the length and diameter of the bolts for segmental connection are about 1000mm and 25mm, respectively. Meanwhile, both the length and diameter of the first segmental connection are 1000mm and 32mm, respectively. Each segment is arranged with eight tendons whereas each connection is arranged with eight holes.

## 3 The outlines design for communication tower

In this research, a distinct tall slender monopole communication tower with 30m height, and 16 Tons weight is considered for design and construction purpose. Both in-situ and pre-cast construction methods were suitable for communication tower construction. To design the tower, it is preferable to rely on the precast construction system to essay casting of the tower segment that quickens the construction process. Fig 1 illustrates three numbers of 10m long precast tapered-tubular segments within the tower. Bolts and nuts are used to connect all segments as showed in Fig 2. The length of both bolts and diameter is about 1000mm and 25mm,

**Table 2. Mix design for UHPFRC with micro and hooked ends steel fibers.**

| Material | Weight (kg/m$^3$) |
|---|---|
| Cement | 850 |
| Densified Silica Fume (SF90) | 200 |
| Dry Silica Fine Sand 30/100 PB | 695 |
| Dry Silica Coarse Sand 16/30 PB | 295 |
| Silica VC2644 | 40 |
| Steel Fiber (WSF 0220) | 79 |
| Steel Fiber (C-GSF0325) | 79 |
| Free Water | 140 |
| 3% moisture | 30.93 |
| Total air voids | - |
| Total | 2408.93 |

**Table 3. The material properties of steel reinforcement used.**

| Type | Steel Reinforcement Grade460 (BS4449) | | | | | |
|---|---|---|---|---|---|---|
| | **T10** | **T12** | **T16** | **T20** | **T25** | **T32** |
| Dia. (mm) | 10 | 12 | 16 | 20 | 25 | 32 |
| Density of Steel (kg/m$^3$) | 7840 | 7840 | 7840 | 7840 | 7840 | 7840 |
| $A_s$(mm$^2$) | 79 | 113 | 201 | 314 | 491 | 804 |
| $E_s$(GPa) | 200 | 200 | 200 | 200 | 200 | 200 |
| $\sigma_{sy}$(MPa) | 460 | 460 | 460 | 460 | 460 | 460 |
| $\varepsilon_{sy}$ | 0.002 | 0.002 | 0.002 | 0.002 | 0.002 | 0.002 |
| $F_{sy}$(kN) | 36 | 52 | 92 | 115 | 226 | 370 |

respectively. Meanwhile, the length and diameter of the first segment connection are about 1000mm and 32mm, respectively.

The bottom segment comprises 3.09m$^3$ of UHPFC and weighs 7.42 tons. The intermediate segment comprises2.18m$^3$ of UHPFC and weighed 5.32 tons. The top segment comprises 1.35m$^3$ of UHPFC and weighed 3.24 tons. Design provisions are given in EN Code. [27–29]. Table 6 shows the segmental and connection details.

When segments become thicker and the fabrication issues are costly, using hollow segmental can reduce the weight and the cost of transportation and construction. The segments are utilized by internal prestressing tendon with a diameter of 15.2mm. Each segment is arranged with eight tendons whereas each connection is arranged with eight holes for bolts installation. Table 7 listed the properties of prestressed strands.

The tower is fixed to a reinforcement concrete foundation block with the dimensions of 4.00m×4.00m and 1m deep. High strength bolts and nuts are used to connect the tubular UHPFC precast segments located at both end edge of the segments.

**Table 4. Material properties of UHPFC.**

| Item | At transfer | At service |
|---|---|---|
| | **(t = 2 days)** | **(t = infinity)** |
| Characteristic Cube Compressive Strength, f$_{Ucu}$ (MPa) | 85 | 165 |
| Characteristic Cyl. Compressive Strength, f$_{Ucu}$ (MPa) | 80 | 160 |
| Characteristic Compressive Strength, f$_{Uck}$ (MPa) | 70 | 150 |
| Density of Concrete (kg/m$^3$) | 2450 | 2450 |
| Characteristic Elastic Tensile Strength, f$_{U,tek}$ (MPa) | 40 (at joint) | 70 (at joint) |
| Characteristic Ultimate Tensile Strength, f$_{U,tuk}$ (MPa) | 40 (at joint) | 70 (at joint) |
| Modulus of Elasticity, E$_U$ (GPa) | 40 | 50 |
| Poisson's Ratio | 0.2 | 0.2 |

**Table 5. Mechanical properties of UHPFC material.**

| Age (days) | Maximum compressive stress | Corresponding strain |
|---|---|---|
| | **(MPa)** | **(µε)** |
| 7 | 211.27 | 2394 |
| 14 | 219.38 | 5328 |
| 28 | 242.61 | 6897 |

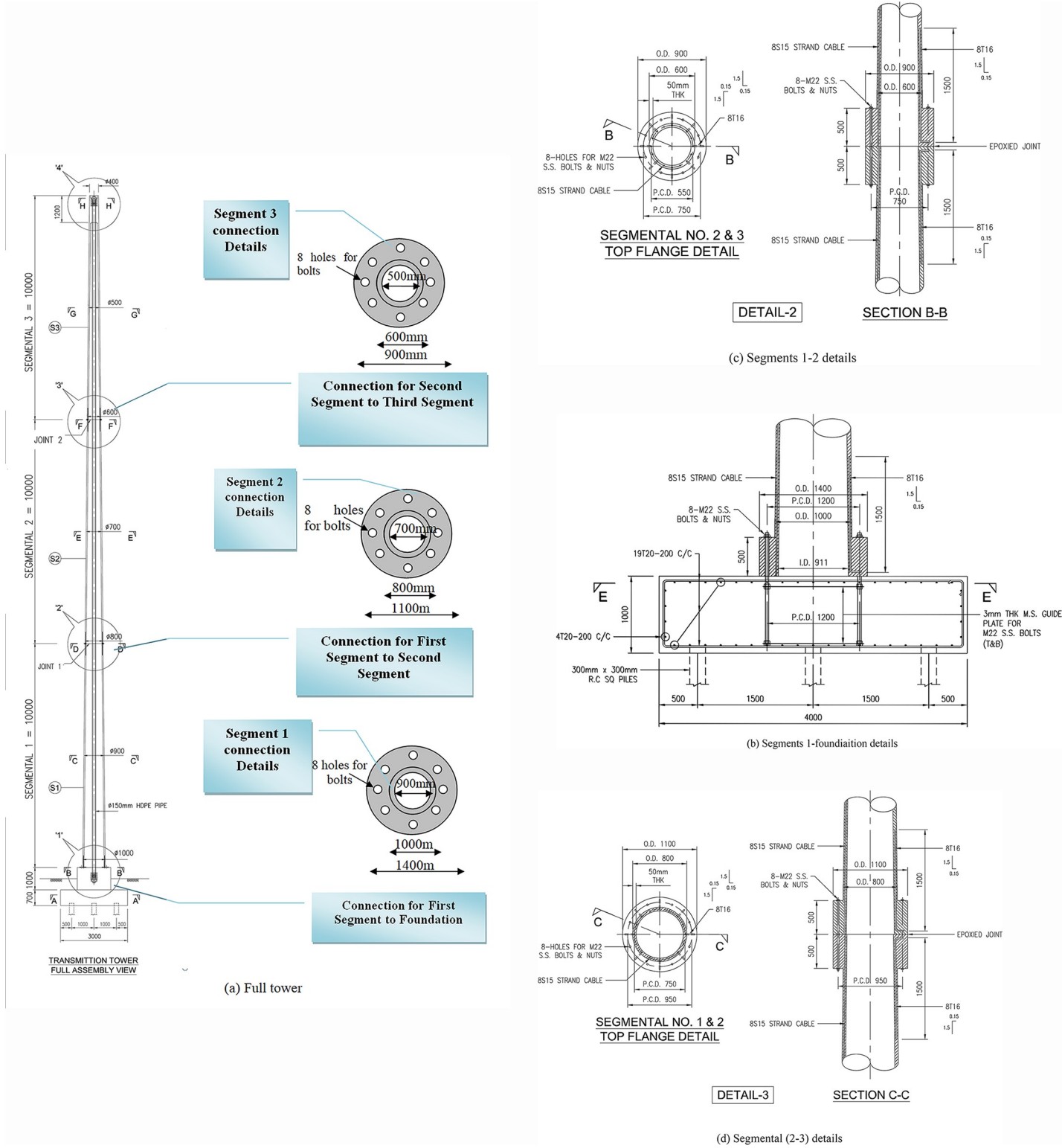

**Fig 1. Local detailed diagram.** (a) Full tower. (b) Segments 1-foundiaition details. (c) Segments 1–2 details. (d) Segmental (2–3) details.

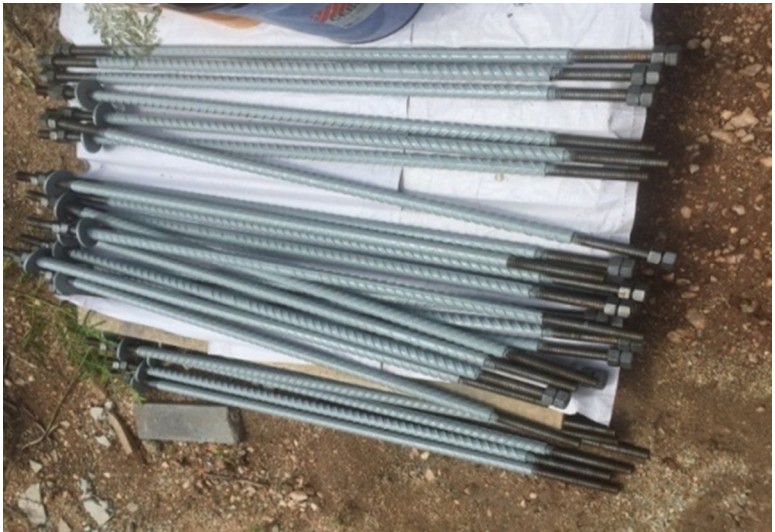

**Fig 2. Screw and nut for tower connection.**

## 4 Design detail for UHPFC communication tower

UHPFC tower segments are pretension precast tapered segments made from steel fiber reinforced UHPFC, thus transformed sectional properties can be calculated as follow:

The section modulus ratios for strands are taken as $n_p = E_p / E_c$. The $n$ values of transfer and service are determined from Eq (1):

$$n_{transfer} = 195/40 = 4.88 n_{service} = 195/50 = 3.9 \tag{1}$$

Table 8 listed the details of the Transformed Section Properties of segments at transfer and service stage.

### 4.1 At transfer stage

Assume all the strands prestress 75% of their breaking load (i.e. 260 kN) with a loss of 5% during transfer. However, there is a 10% of the long term losses after the segments have undergone all time-effect losses such as creep and shrinkage and relaxation of the tendons.

**Table 6. Tower details.**

| Type | Height | External diameter at bottom | Internal diameter at bottom | External diameterat top | Internal diameter at top | Thickness |
|---|---|---|---|---|---|---|
| | (m) | (mm) | (mm) | (mm) | (mm) | (mm) |
| Segmental 1 | 10 | 1000 | 900 | 800 | 700 | 50 |
| Segmental 2 | 10 | 800 | 700 | 600 | 500 | 50 |
| Segmental 3 | 10 | 600 | 500 | 400 | 300 | 50 |
| Connection for segmental 1 to foundation | 0.5 | -- | -- | 1400 | 900 | 500 |
| connection for segment 1 to 2 (upper and lower) | 0.5 | 1100 | 800 | 1100 | 700 | 400 |
| connection for segment 2 to 3 (upper and lower) | 0.5 | 900 | 600 | 900 | 500 | 400 |

Table 7. The material properties of prestressed strand used.

| Type | S15 |
| --- | --- |
| Diameter (mm) | 15.2 |
| Density of Steel (kg/m$^3$) | 7840 |
| Type | Low Relaxation |
| Nominal Section (mm$^2$) | 140 |
| Nominal Weight (kg/m) | 1.10 |
| Specified Breaking Load (kN) | 260 |
| Specified Load at 1% Elongation (kN) | 235 |
| Modulus of Elasticity, $E_p$ (GPa) | 195 |
| $n_p$ (section modulus ratios) | 4.88 (at transfer) |
| | 3.90 (at service) |

The jacking force for each strand can be obtained from Eq (2):

$$P_j = 260 \text{ kN x } 0.75 = 195 \text{ kN} \tag{2}$$

The initial prestressed force can be calculated from Eq (3):

$$P_i = 0.95 \, P_j = 0.95 \text{ x } 195 = 185.25 \text{ kN (used during transfer, } t = 2 \text{ days)} \tag{3}$$

The effective prestressed force is concluded through Eq (4):

$$P_e = 0.90 \text{ x } P_i = 0.9 \text{ x } 185.25 = 167 \text{ kN (used during service, } t = \text{infinity)} \tag{4}$$

The extreme fiber stresses during transfer within the support regions (ends of segment) are:

- The resulta stress at the top of the segment is -9.73 MPa in compression, which is less than the stress limit of the concrete in compression during transfer.

- The resultant stress at the bottom of the segment is -9.73 MPa in compression, which far less than the stress limit of concrete in compression. In short, prestressed UHFPC segments have a lot of reserve compression capacities. The crush in the concrete will never occur. The segment will not crack during the transfer of the prestress.

## 4.2 At service stage

Assume all the strands prestress 75% of their breaking load (i.e. 260 kN) with a loss of 5% during transfer. However, there is a 10% of long-term losses after the segments have experienced all time-effect losses such as creep and shrinkage.

Table 8. Details of segments at transfer and service.

| Item | At transfer (t = 2 days) | At service (t = infinity) |
| --- | --- | --- |
| n | 4.33 | 3.9 |
| A (mm$^2$) | 152,300 | 151,100 |
| $I_{xx}$ (x 10$^9$ mm$^4$) | 17.02 | 16.89 |
| $y_t$ (mm) | 500 | 500 |
| $y_b$ (mm) | 500 | 500 |
| $Z_t$ (x 10$^6$ mm$^3$) | 34.04 | 33.79 |
| $Z_b$ (x 10$^6$ mm$^3$) | 34.04 | 33.79 |

The jacking force for each strand is given by Eq (5):

$$P_j = 260 \text{ kN x } 0.75 = 195 \text{ kN} \tag{5}$$

The initial prestressed force can be defined as:

$$P_i = 0.95 \, P_j = 0.95 \text{ x } 195 = 185.25 \text{ kN (used during transfer, } t = 28 \text{ days)} \tag{6}$$

The effective prestressed force can be expressed as:

$$P_e = 0.90 \text{ x } P_i = 0.90 \text{ x } 185.25 = 167 \text{ kN (used during service, } t = \text{infinity)} \tag{7}$$

The extreme fiber stresses during service within the support regions (ends of segments) are:

- The resultant stress at the top of the segment is -9.73 MPa in compression, which is less than the stress limit of the concrete in compression during transfer.

- The resultant stress at the bottom segment is -8.84 MPa in compression, which is far less than the stress limit of concrete in compression. In other words, prestressed UHFPC segments have a lot of reserve compression capacities, hence the crush of the concrete will never occur.

### 4.3 Design Moment Resistance (DMR)

The design moment resistance can be obtained using the moment equilibrium method. The first factored design tendon tension forces are calculated. The material of the strands is 1.15 and 1.3 for UHPFC. The factored compressive stress can be determined from Eq (8):

$$f_{Uck}/\gamma = 150/1.3 = 115 \text{ MPa. Response2000 gives a value of } M_{Rd} = 838 \text{ kNm} \tag{8}$$

The below sections describe various components of the tower.

## 5 Construction procedure of tubular precast segment

In this research, an advanced construction technique is developed to construct UHPFC communication tower. The concrete construction costs heavily depend on the methods used to achieve performance.

The parts of the tower are manufactured as precast units, thus high quality and short processing time can be achieved. Concurrently, prefabricated concrete units are assembled by crane on top of another and tied with the bolts and nuts.

The sections below discuss the developed construction process for various components of the tower.

### i. Step 1: Foundation system

The tower is founded with high performance reinforced concrete pile to support the raft foundation. The reinforced concrete raft foundation utilized high-performance Self Compacting Concrete (SCC). Fig 3 shows the Raft Foundation System. The diameter of 1500mm and height of 18m reinforced concrete bored piles are set to construct the foundation of the tower below the base of the raft. The capacity of the assumed pile is 3000tonnes. A robust catholic protection system issued for both the bored piles and raft foundation against the severe and corrosive environment (chloride and sulfate) of the soil.

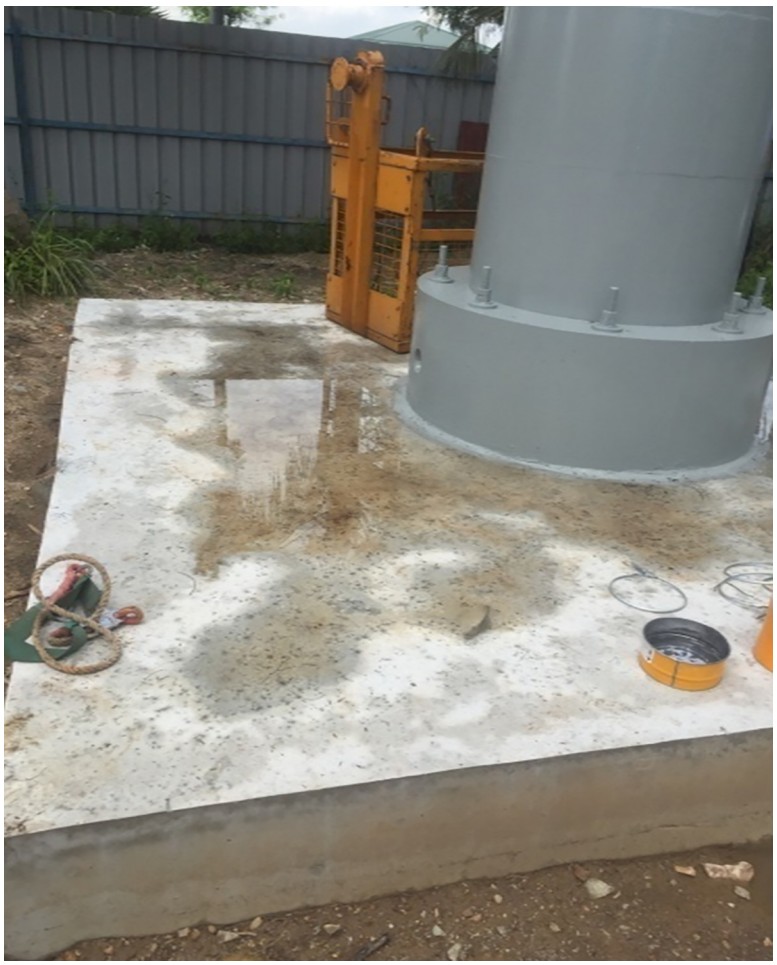

**Fig 3. Tower foundation.**

### ii. Step 2: Fabricated the moulds

The mould and precast segments are fabricated in Dura Technology Sdn. Bhd., located in Ipoh, Perak, Malaysia. In order to cast tubular concrete segments, both internal and external moulds are fabricated (Fig 4). During concrete casting, the internal mould is placed inside the external mould. After that, the internal mould is removed out of the segment.

### iii. Step 3: Concreting

The process of concrete for each segmental members was a challenging issue due to hollow section configuration, high length of each segment (10m) and positioning of process. In this study, a new process of concreting segmental tower is implemented as described below

- Crane and chain are used to locate the inner steel mould inside the outer mould. Then, the inner mould is positioned.

- Hydraulic jack is used to present the tendons.

The mould is fixed vertically as support structure (see Fig 5). The support structure is designed and fabricated to hold the mold in vertical condition. After fixing the inner and

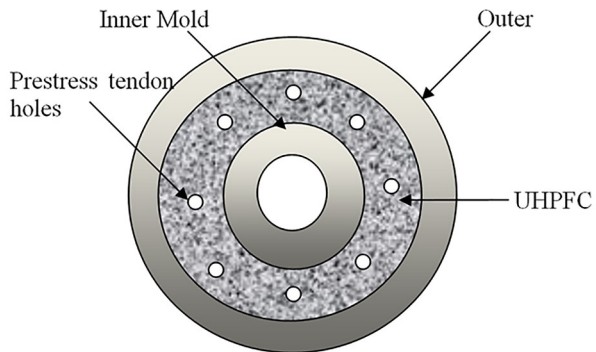

(a) Mold section

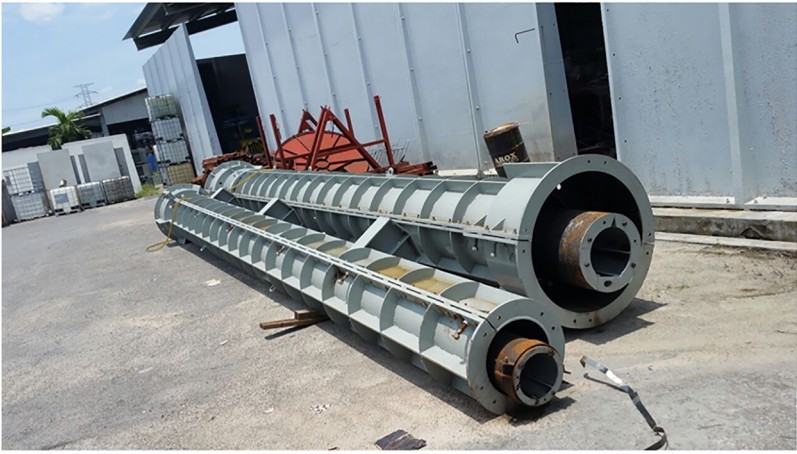

(b) Steel mold for casting tower segmental and connection

**Fig 4. Steel mold details.** (a) Mold section. (b) Steel mold for casting tower segmental and connection.

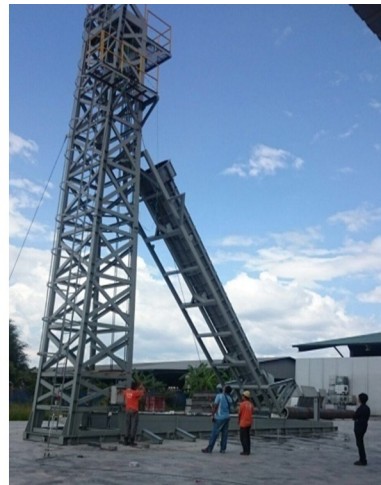 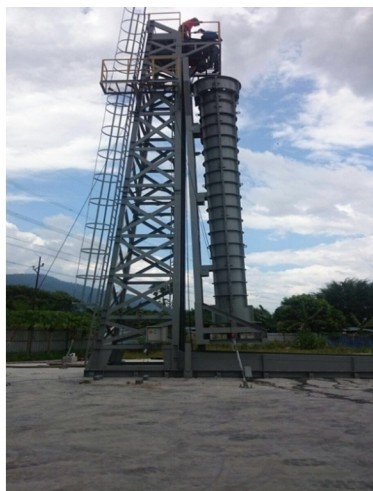

(a) Mould pushing to vertical condition    (b) Mould in vertical position for concreting

**Fig 5. Fixing the mould inside the support structure.** (a) Mould pushing to vertical condition. (b)Mould in vertical position for concreting.

outer mould, cables are used to rotate the vertical condition in the support tower for concreting purpose. Once the concrete is cured, the mould returns to the horizontal position to remove the molds.

As shown in Fig 6, the crane is used to deposit the concrete vertically in hollow cylinder mould to cast the first, second and third segmental. The concrete vibration is carried out during concreting by using the vibrator, installed at the top of the mold (refer to Fig 7).

After curing the concrete segments, the mould is rotated to horizontal condition. Then, jack and chain are used to remove the inner mould as illustrated in Fig 8.

The last step is to remove the outer mould and continue curing the concrete.

### iv. Step 4: Pre-stressing

Based on Fig 9, hydraulic jack with initial prestress 1200Mpa is used to post-tension the eight tendons for each segment.

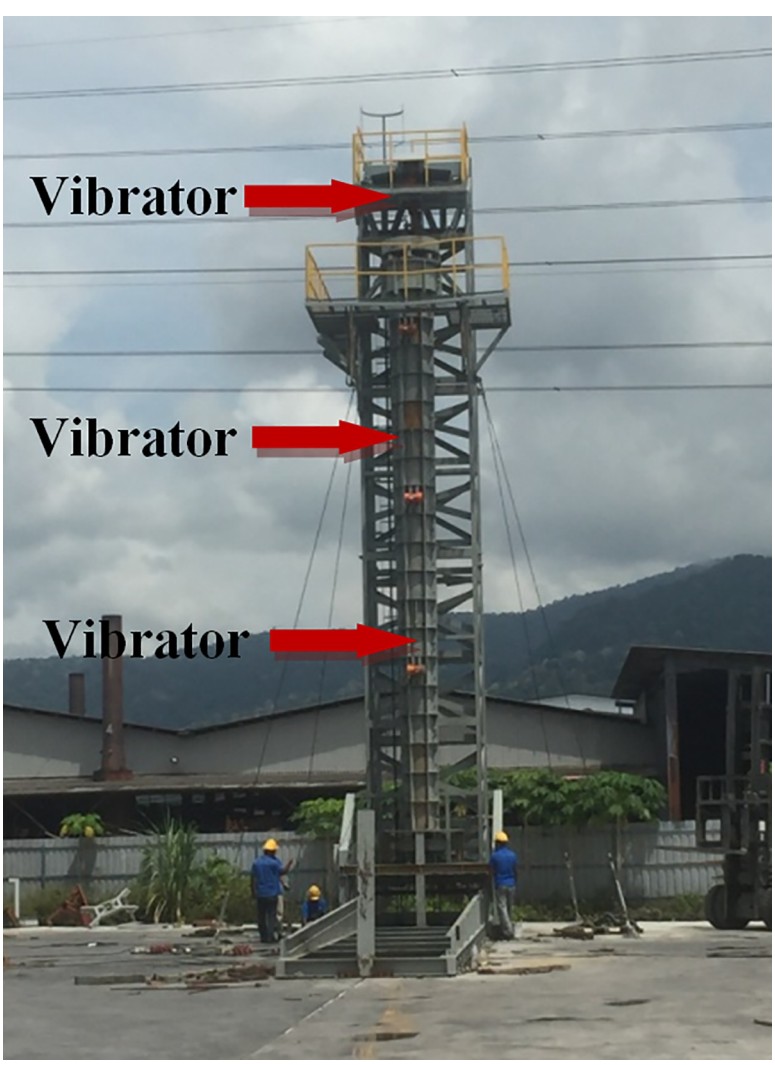

**Fig 6. Using vibrators for concrete mixing.**

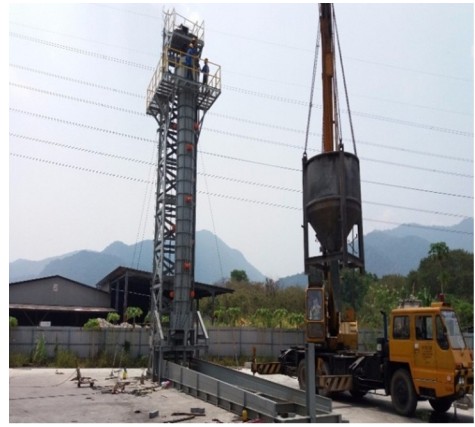 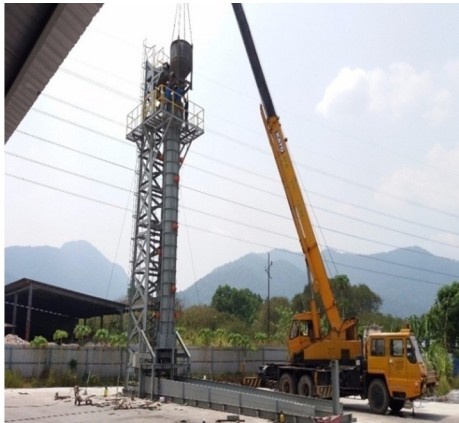

(a) Prepare to deposit the concrete          (b) deposit concrete through the mould

**Fig 7. UHPFC deposit concrete for segmental casting.** (a) prepare to deposit the concrete. (b) deposit concrete through the mould.

### v. Step 5: Making eight holes at connection to locate the bolts place

In the second stage of construction, the bolts are used to connect different segments. Eight holes are made to connect each segment. They are positioned equally for each segmental connection in order to locate the bolts in the same positions as depicted in Fig 10.

Before casting begins, it is crucial to apply resin around the area of the inner mould. Talcum powder can also be used as a release agent during the casting process.

### vi. Step 6: Installation procedure

After the construction of the tower foundation system is complete, bolts and nuts are used during the installation process to join the segments as showed in Fig 11. An epoxy layer is

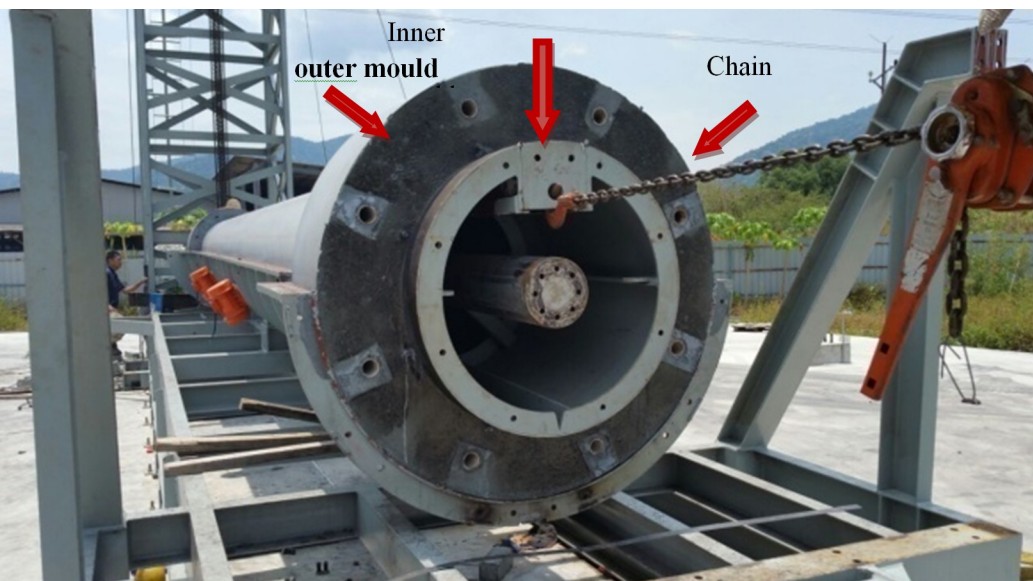

**Fig 8. Removing the moulds from concrete segments.**

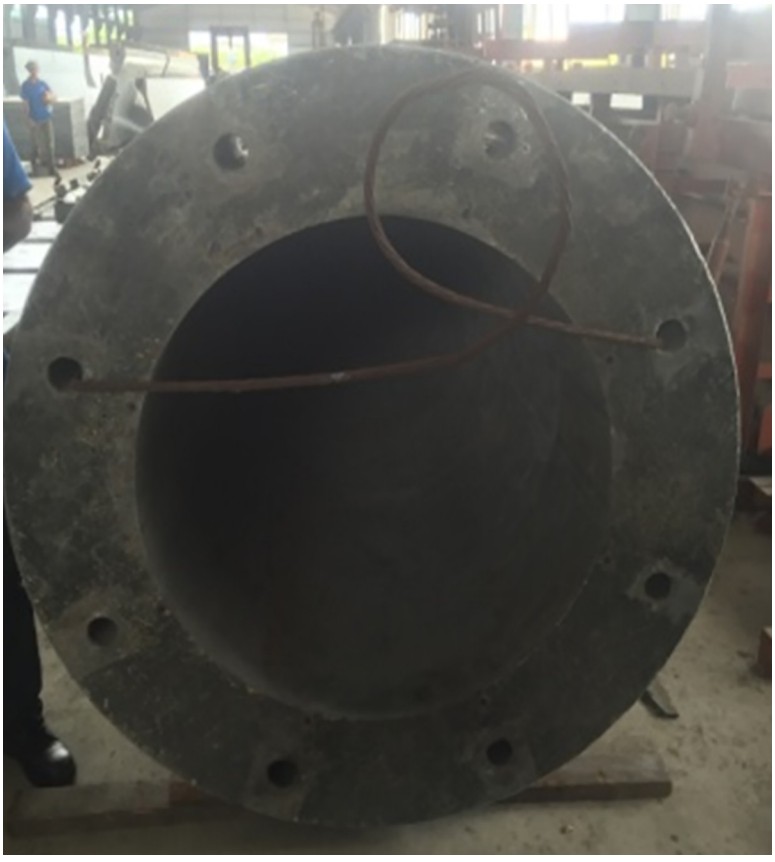

**Fig 9. Prestressing.**

also used in the interface between concrete segmental connections. A 25-ton self-climbing luffing type tower crane is located beside the tower. It is used to lift the precast concrete segments.

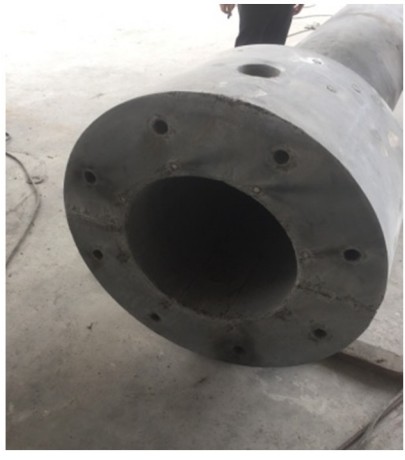
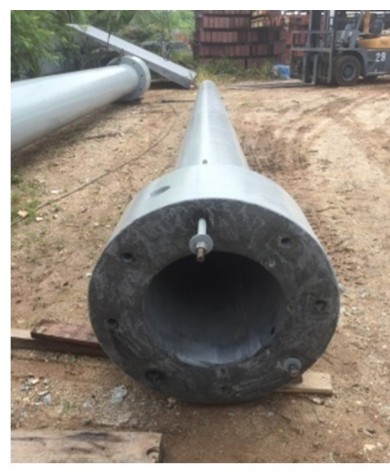

(a) Create eight holes to connect each segment     (b) Locate the bolts in the holes

**Fig 10. Location for bolts holes.** (a) Create eight holes to connect each segment. (b) locate the bolts in the holes.

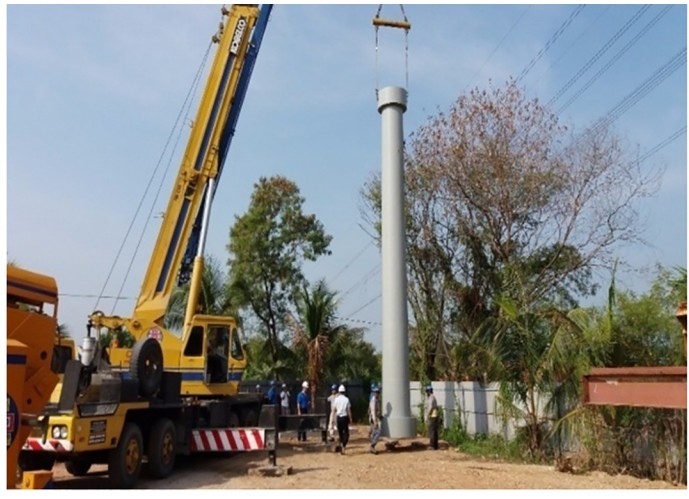

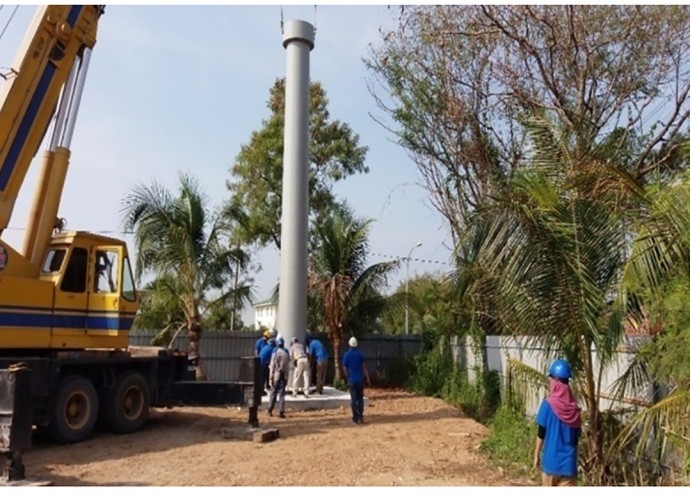

(a) Fetch the segment 1                    (b) locate segment 1 on the foundation

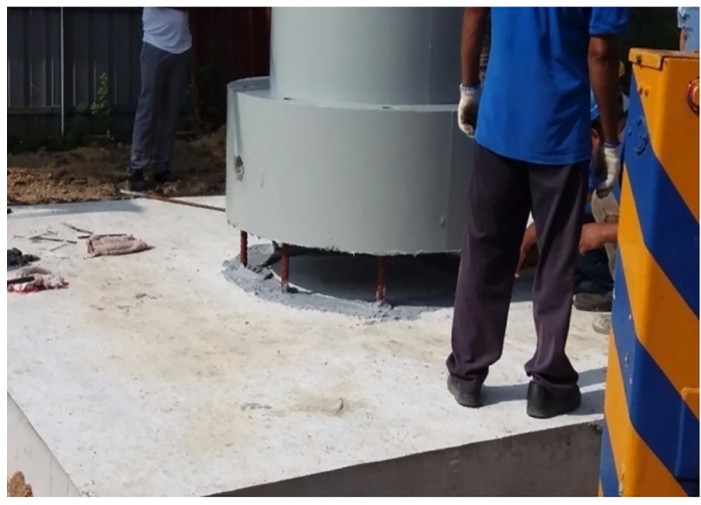

(c) Locate segment 1 on the foundation          (d) tie segment1 to the foundation

**Fig 11. Shows the installation of the first segment.** (a) Fetch the segment 1. (b) locate segment 1 on the foundation. (c) locate segment 1 on the foundation. (d) tie segment1 to the foundation.

Then left of segmental two above the segmental one by using the crane and connect them by using bolts and nuts with 5mm of the epoxy layer (see Fig 12).

After completing the installation of segmental two, it is followed by the segmental three with the assistance of the crane above segmental two. Bolts and nuts with 5mm of epoxy layer were used to connect these segments (see Fig 13).

Fig 14 shows the tower after the installation of tower segments is complete.

### vii. Step 7: Operating of the tower

Most projects are complete as soon as the installation works are finished and the system operates. Fig 15 shows the installation of a staircase in the tower.

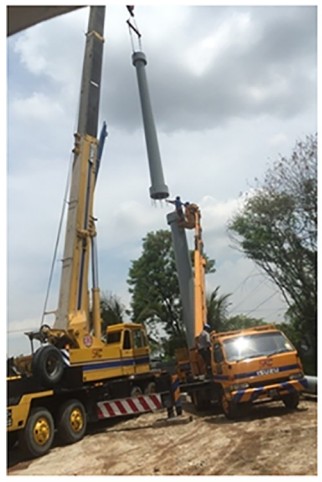 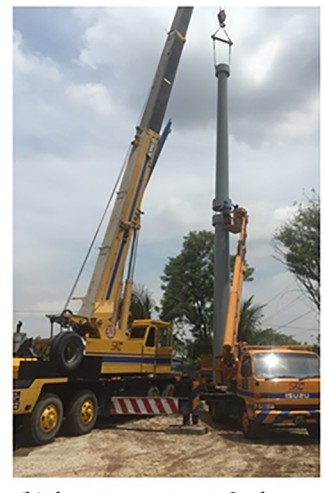 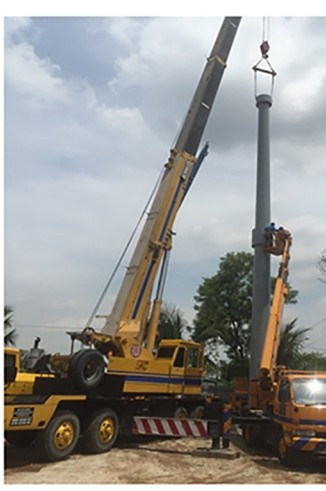

(a) Fetch the segment 2      (b) locate segment 2 above      (c) tie segment 2 to 1
                                   of segment 1

**Fig 12. Segmental two installations.** (a) Fetch the segment 2. (b) locate segment 2 above of segment1 (c) tie segment 2 to 1.

## 6 Conclusion

The technology development allows communication tower application to grow rapidly. Tower typology varies across countries in structural design. This paper highlights the new design and construction method for UHPFC communication tower. In this study, the construction technique is developed to build the UHPFC communication tower according to the precast system implementation for the body of the tower with three segments. These segments are manufactured as precast segments to achieve high quality and short processing time. Prefabricated UHFPC concrete units are assembled by crane, which is tied with bolts and nuts. The advantage of using precast concrete system is it is the fastest construction process on site. Using hollow segmental parts can reduce the weight of the tower, transportation cost, and construction procedure. Moreover, implementing high-strength steel strands post-tension concrete tower

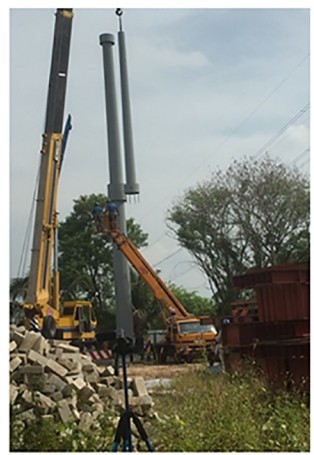 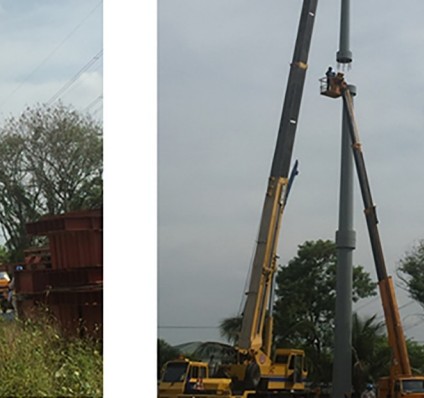 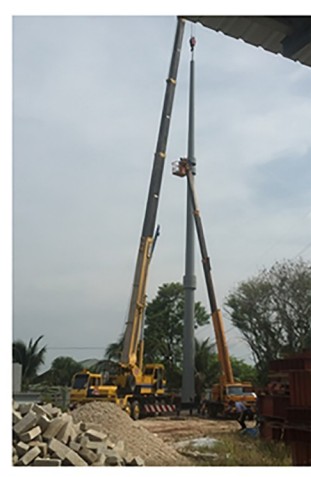

(a) Fetch the segment 3      (b) locate segment 3 on segment 2      (c) tie segment 3 to 2

**Fig 13. Segmental three installations.** (a) Fetch the segment 3. (b) locate segment 3 on segment 2. (c) tie segment 3 to 2.

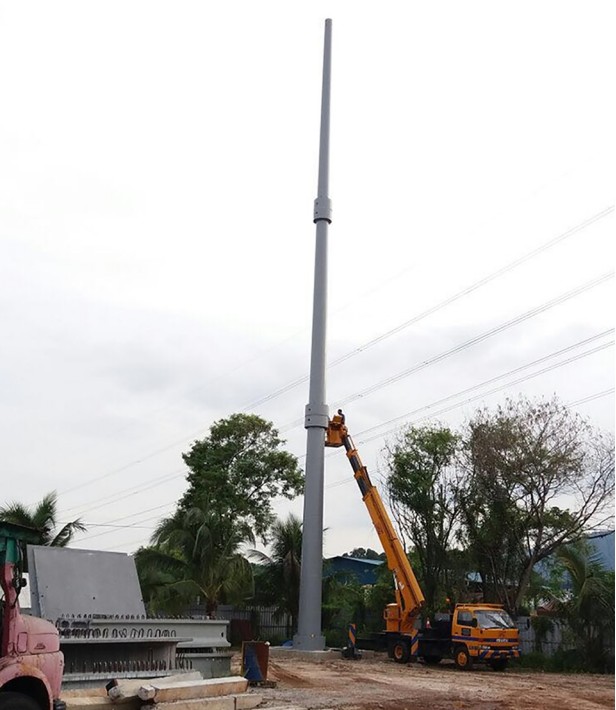

**Fig 14. Complete tower installation.**

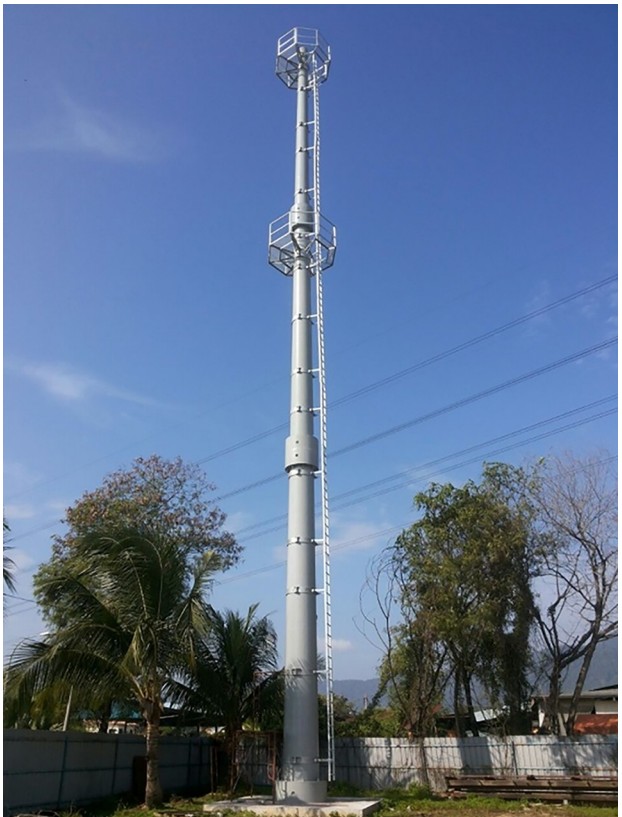

**Fig 15. Tower after finishing the construction and installation of stair case.**

can increase the structure's bending strength, shear strength, and resistance to lateral load. The proposed design is applicable for various grounds in different climate conditions to facilitate the construction and installation procedure.

## Supporting information

**S1 File.**
(PDF)

## Author Contributions

**Conceptualization:** Voo Yen Lai.

**Data curation:** Farzad Hejazi, Sarah Saleem.

**Formal analysis:** Farzad Hejazi.

**Funding acquisition:** Voo Yen Lai.

**Investigation:** Sarah Saleem.

**Methodology:** Voo Yen Lai.

**Project administration:** Farzad Hejazi.

**Resources:** Voo Yen Lai.

**Supervision:** Farzad Hejazi.

**Validation:** Farzad Hejazi.

**Visualization:** Sarah Saleem.

**Writing – original draft:** Farzad Hejazi.

**Writing – review & editing:** Voo Yen Lai.

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
