## [Decision Letter · Decision Letter 0]

27 May 2020

PONE-D-20-10814

The construction process for pre-stressed ultra high performance concrete communication tower

PLOS ONE

Dear Dr. Hejazi,

Thank you for submitting your manuscript to PLOS ONE. After careful consideration, we feel that it has merit but does not fully meet PLOS ONE’s publication criteria as it currently stands. Therefore, we invite you to submit a revised version of the manuscript that addresses the points raised during the review process.

We look forward to receiving your revised manuscript.

Kind regards,

Martin Koller, Ph.D.

Academic Editor

PLOS ONE

Journal Requirements:

'This research receives support from Dura Technology Sdn. Bhd. (https://dura.com.my)

via Dr. Voo Yen Lai with grant project with Vot No. 6300195.

Dura Technology Sdn. Bhd. received this fund from Ministry of Science Technology &

Innovation, Malaysia (https://www.mestecc.gov.my) for the project entitled as

“Development and construction of Internet Transmission Tower Using Ultra High

Performance Concrete”.

Their help and support are gratefully acknowledged.

The sponsors or funders have no any role in the study design, data collection and

analysis, decision to publish, or preparation of the manuscript.'

We note that you received funding from a commercial source: Dura Technology Sdn. Bhd

4. Please ensure that you refer to Figures 2 and 11 in your text as, if accepted, production will need this reference to link the reader to the figure.

Reviewers' comments:

Reviewer's Responses to Questions

**Comments to the Author**

1. Is the manuscript technically sound, and do the data support the conclusions?

Reviewer #1: Yes

Reviewer #2: Yes

2. Has the statistical analysis been performed appropriately and rigorously? 

Reviewer #1: N/A

Reviewer #2: Yes

3. Have the authors made all data underlying the findings in their manuscript fully available?

Reviewer #1: Yes

Reviewer #2: Yes

4. Is the manuscript presented in an intelligible fashion and written in standard English?

Reviewer #1: No

Reviewer #2: Yes

5. Review Comments to the Author

Reviewer #1: The submitted manuscript very interesting in presentation of new design for construction of segmental tubular section communication tower with ultra-high-performance concrete (UHPFC) material and prestress tendon to gain durability, ductility, and strength.

1. The composition of the Ultra high-performance concrete (UHPFC) has to recognized in the abstract and inside the article.

2. Unify the unities used in the whole manuscript.

3. The workload of this paper is huge, and it has achieved relatively significant results, but the experimental level should be strengthened. It is important to provide with chemical composition and physical characterization of (UHPFC).

4. A thorough English language editing is required.

5. The authors can provide the introduction and background about other lower performance concrete and cementitious composites compared to the used (UHPFC) relative to the durability, ductility, and strength; some literatures in concern:

• H.M. Saleh, M.E. Tawfik and T.A. Bayoumi, Chemical stability of seven years aged cement–PET composite waste form containing radioactive borate waste simulates, Journal of Nuclear Materials, 411(1-3), 185-192, (2011).

• S.B. Eskander T.A. Bayoumi and H.M. Saleh, Performance of aged cement-polymer composite immobilizing borate waste simulates during flooding scenarios, Journal of Nuclear Materials 420(1-3), 175-181, (2012).

• S.B. Eskander and H.M. Saleh, Cement mortar-degraded spinney waste composite as a matrix for immobilizing some low and intermediate level radioactive wastes: Consistency under frost attack, Journal of Nuclear Materials 420(1-3), 491-496, (2012).

• H.M. Saleh and S.B. Eskander, Characterizations of mortar-degraded spinney waste composite nominated as solidifying agent for radwastes due to immersion processes, Journal of Nuclear Materials, 430(1-3), 106-113, (2012).

• H.M. Saleh, S.B. Eskander and H.M. Fahmy, Mortar composite based on wet oxidative degraded cellulosic spinney waste fibers, International Journal of Environmental Science and Technology, 11(5), 1297-1304, (2014).

• H.M. Saleh, F.A. El-Saied T.A. Salaheldin and A.A. Hezo, Macro- and Nanomaterials for Improvement of Mechanical and Physical Properties of Cement Kiln Dust-Based Composite Materials. Journal of Cleaner Production 204, 532-541, (2018).

• H.M. Saleh, S.M. El-Sheikh, E.E. Elshereafy and A.K. Essa, Mechanical and physical characterization of cement reinforced by iron slag and titanate nanofibers to produce advanced containment for radioactive waste. Construction and Building Materials 200(C), 135-145, (2019).

• H.M. Saleh, F.A. El-Saied T.A. Salaheldin and A.A. Hezo, Influence of severe climatic variability on the structural, mechanical and chemical stability of cement kiln dust-slag-nanosilica composite used forradwaste solidification. Construction and Building Materials 218(C), 556-567, (2019).

• H.M. Saleh and S.B. Eskander, Impact of water flooding on hard cement-recycled polystyrene composite immobilizing radioactive sulfate waste simulate. Construction and Building Materials 222C, 522-530, (2019).

• H.M. Saleh, S.M. El-Sheikh, E.E. Elshereafy and A.K. Essa, Performance of cement-slag-titanate nanofibers composite immobilized radioactive waste solution through frost and flooding events. Construction and Building Materials 223C, 221-232, (2019).

Reviewer #2: Authors have conducted a preliminary research on The construction process for pre-stressed ultra high performance concrete communication tower. While the topic will be of interest to readers of PLOS ONE, there are several issues which the authors must address for the article to be up to the standard of this journal

Add the mix design of ultra high performance concrete.

make an appropriate contribution sentence.

bold the novelty

what is objective,problem statement and knowledge gap?

6. PLOS authors have the option to publish the peer review history of their article (what does this mean?). If published, this will include your full peer review and any attached files.

Reviewer #1: No

Reviewer #2: No

---

## [Author Response · Author response to Decision Letter 0]

14 Aug 2020

PONE-D-20-10814R1

Title: The construction process for pre-stressed ultra high performance concrete communication tower

Required Correction 1) 

Please ensure that you refer to Table 4,5 and 7 in your text as, if accepted, production will need this reference to link the reader to the Table.

Author reply:

All tables are cited in the manuscript.

Required Correction 2) 

We note that you currently have two Tables in your manuscript titled as Table 1 and two Tables titled as Table 2.

The first "Table 1: Mix design for UHPFRC used for construction of communication tower - and the second "Table 1: Mix Design for UHPFRC With Micro and Hooked Ends Steel Fibers.

The first "Table 2: The material properties of steel reinforcement used - and the second "Table 2: Mechanical properties of UHPFC material. So that these tables can be differentiated can you please update the Table title numbering and the in-text citations to them accordingly.

Author reply:

Thanks for the comment and apologize for the typo mistakes. All table numbers are sorted carefully. 

Required Correction 3) 

Based on the information you have provided us, we propose the following funding and competing interests statements for your approval:

Funding:

"This research receives support from Dura Technology Sdn. Bhd. (https://dura.com.my) via Dr. Voo Yen Lai with grant project with Vot No. 6300195. Dura Technology Sdn. Bhd. received this fund from Ministry of Science Technology & Innovation, Malaysia (https://www.mestecc.gov.my) for the project entitled as “Development and construction of Internet Transmission Tower Using Ultra High Performance Concrete”. Their help and support are gratefully acknowledged. The sponsors or funders have no any role in the study design, data collection and analysis, decision to publish, or preparation of the manuscript."

Competing interests:

This research receives support from Dura Technology Sdn. Bhd. (https://dura.com.my). This does not alter our adherence to PLOS ONE policies on sharing data and materials.

Can you please confirm the above statement is both complete and correct? With your approval we will update these statements on your behalf. If you approve of them, please make this clear in your cover letter or author comments.

Author reply:

Yes, I am confirmed the above statements. This statements are added in the cover letter. 

Required Correction 4) We note your current Data Availability statement: "The data underlying the results presented in the study are available from Corresponding Author."

Please note that PLOS journals require authors to make all minimal data underlying the findings described in their manuscript fully available without restriction at the time of publication, with rare ethical or legal exceptions: https://journals.plos.org/plosone/s/data-availability.

PLOS ONE describes the minimal data set as that which is used to reach the conclusions drawn in the manuscript with related metadata and methods, and any additional data required to replicate the reported study findings in their entirety. This may include:

- The values behind the means, standard deviations and other measures reported

- The values used to build graphs

- The points extracted from images for analysis

Please provide the name of the ethics committee or Institutional Review Board that is imposing sharing restrictions on your de-identified minimal data set and the grounds for restriction (e.g. contains sensitive identifying information).

Author reply:

There is no restriction on presented data in this paper and all data already provided within manuscript.

---

## [Editor Report · Decision Letter 1]

21 Aug 2020

The construction process for pre-stressed ultra high performance concrete communication tower

PONE-D-20-10814R1

Dear Dr. Farzad Hejazi,

We’re pleased to inform you that your manuscript has been judged scientifically suitable for publication and will be formally accepted for publication once it meets all outstanding technical requirements.

Kind regards,

Martin Koller, Ph.D.

Academic Editor

PLOS ONE
---

## [Editor Report · Acceptance letter]

9 Oct 2020

PONE-D-20-10814R1 

The Construction Process For Pre-Stressed Ultra High Performance Concrete Communication Tower 

Dear Dr. Hejazi:

I'm pleased to inform you that your manuscript has been deemed suitable for publication in PLOS ONE. Congratulations! Your manuscript is now with our production department. 

Kind regards, 

on behalf of

Dr. Martin Koller 

Academic Editor

PLOS ONE